# Transdermal Maltose-Based Microneedle Patch as Adjunct to Enhance Topical Anesthetic before Intravenous Cannulation of Pediatric Thalassemic Patients Receiving Blood Transfusion: A Randomized Controlled Trial Protocol

**DOI:** 10.3390/jcm11185291

**Published:** 2022-09-08

**Authors:** Muhammad Irfan Abdul Jalal, Kai Shen Ooi, Kai Cheong Foo, Azrul Azlan Hamzah, Chee Seong Goh, Chang Fu Dee, Poh Choon Ooi, Muhamad Ramdzan Buyong, Teck Yew Low, Xin Yun Chua, Doris Sie Chong Lau, Zarina Abdul Latiff, Fook Choe Cheah

**Affiliations:** 1UKM Medical Molecular Biology Institute, Universiti Kebangsaan Malaysia Medical Centre, Kuala Lumpur 56000, Malaysia; 2Department of Paediatrics, Faculty of Medicine, Universiti Kebangsaan Malaysia Medical Centre, Kuala Lumpur 56000, Malaysia; 3Institute of Microengineering and Nanoelectronics, Universiti Kebangsaan Malaysia, Bangi 43600, Malaysia; 4Alnair Labs Corporation, Tokyo 141-0031, Japan; 5Pharmacy Department, Universiti Kebangsaan Malaysia Medical Centre, Kuala Lumpur 56000, Malaysia

**Keywords:** thalassemia, skin analgesia, dissolvable, pain, microneedle, VAS scores, skin conductance algesimeter index, randomized phase II cross-over trial

## Abstract

Intravenous cannulation is experientially traumatic to children. To minimize this, EMLA^®^ is applied on the would-be-cannulated area before IV cannula insertion. However, the time to achieve its maximum efficacy may be affected due to incomplete cutaneous absorption and the duration of application. The latter may be a limiting factor in a busy healthcare facility. The usage of dissolvable maltose microneedles may circumvent this problem by introducing micropores that will facilitate EMLA^®^ absorption. A randomized phase II cross-over trial will be conducted to compare the Visual Analogue Scale (VAS) pain scores and skin conductance algesimeter index between 4 different interventions (1 fingertip unit (FTU) of EMLA^®^ with microneedle patch for 30 min before cannulation; 0.5 FTU of EMLA^®^ with microneedle patch for 30 min; 1 FTU of EMLA^®^ with microneedle for 15 min; 1 FTU of EMLA^®^ with sham patch for 30 min). A total of 26 pediatric patients with thalassemia aged between 6 and 18 years old and requiring blood transfusion will be recruited in this trial. During the visits, the VAS scores and skin conductance algesimeter index at venous cannulation will be obtained using the VAS rulers and PainMonitor™ machine, respectively. The trial will commence in August 2021 and is anticipated to end by August 2022.

## 1. Introduction

### 1.1. Research Background

Intravenous (IV) cannulation elicits considerable pain and is regarded as one of the more frequently encountered traumatic experiences among hospitalized children [1]. The distress and fear toward IV cannulation may further be exacerbated in children with chronic diseases who require regular IV cannulation for blood transfusion or administration of medication [2]. The pain perception in the long term may potentially affect psychological well-being and adversely influence personal self-esteem and relationships. 

When it comes to cutaneous drug delivery, the traditional low-cost small-gauge hypodermic needles and topical cream formulation are the most commonly used approaches. However, both have their drawbacks. For the former, the patients do not tolerate it due to significant pain, anxiety, distress, and the associated needle phobia [3,4]. For topical cream formulation, it appears to be a more attractive option. Topical anesthetic creams, for example, EMLA^®^ Cream (lidocaine 2.5% and prilocaine 2.5%), provide a non-invasive and convenient means of administering anesthesia. As the drug passively diffuses across the skin, it results in a slower onset of effective analgesia, around 60 min [5,6]. This delayed onset of full analgesia is attributed to the relatively impermeable stratum corneum (SC) layer, which impedes its absorption transcutaneously and subsequently impairs its bioavailability, thus limiting its use in a busy clinical setting. Skin thickness variation developmentally, or that affected by underlying medical conditions, may also interfere in this process. Therefore, such drug delivery challenges may affect the acceptability and efficacy of transcutaneous drug delivery [7,8]. 

There are also other state-of-the-art modalities that may boost transdermal delivery of local anesthetic agents, such as phonophoresis (utilizing ultrasound to enhance local anesthetic agent penetration through the skin), magnetophoresis (using magnetic field), iontophoresis (utilizing electrical current to boost electrically charged or ionic local anesthetic agent penetration into the deeper layer of the dermis), and jet injection (J-tip) (using pressurized gas to enhance subcutaneous delivery of local anesthetic agents) [9,10,11]. However, the exorbitant prices and intricacies of the devices used in these techniques and training of operators before such complex devices can be handled effectively are major deterring factors [12].

With the advancement in microfabrication manufacturing technology, research on microneedles (MNs) has been carried out extensively. An MN is a mimic of hypodermic needle composed of hundreds of micron-sized, out-of-plane protrusions, typically arranged in arrays on a patch that can be applied cutaneously [13,14]. MNs create multiple transient microchannels across the skin and disrupt the SC in a minimally invasive manner. These micron-sized conduits augment the skin permeability for better active pharmaceutical ingredient (API) transport into the deeper layers, thus enhancing their therapeutic effects. Furthermore, the MN specification can be adjusted to avoid stimulating the nociceptors and blood vessel penetration. The MN can be fabricated from a variety of materials. Remarkably, the recent focus on biocompatible polymers makes them attractive alternatives over the non-biocompatible silicon or metal-based materials. Microneedle biocompatibility is one aspect to consider in the pediatric population, as the skin barrier function is immature and rapidly evolving [14]. A biocompatible MN eliminates the risk of biohazard waste and possible contaminations if needle breakage occurs. Previous studies have also shown that the microneedle technology for drug delivery is safe, has minimal adverse effects and minimizes the risk of infection compared to the conventional needle system [15,16,17]. Besides, maltose has also been shown to be biocompatible and acceptable polymers for use in humans, making it an ideal candidate for microneedle constituents [18].

So far, there are only three prior published studies investigating the use of microneedles in pediatric settings. Gupta et al. (2011) compared the use of a single hollow microneedle as a means of intradermal delivery of insulin with the subcutaneous administration of insulin using a catheter [19]. They found no significant differences in terms of the area under the curve (AUC) for insulin concentration between those two modalities but faster absorption of insulin using the former. They also found that pain was substantially lower in the MN + insulin group. However, a study by Norman et al. (2013) demonstrated higher pain scores in children aged between 10 and 18 years old who received insulin via MN than their counterparts who received insulin subcutaneously via insulin pump [20]. Nevertheless, the difference was not statistically significant. On the contrary, in a recently published study by Samant et al. (2020) investigating the safety and suitability of different glucose-monitoring modalities in 15 pediatric patients with type I diabetes, the authors demonstrated similar VAS scores between those whose glucose levels were monitored via interstitial fluid (ISF) sampling using MN patch (MN + ISF) and fingerstick capillary blood glucose monitoring [21]. Interestingly, the MN + ISF group reported a statistically significant lower VAS score mean than those on blood glucose monitoring using intravenous catheters [21].

With respect to dissolving microneedle technology, there was a phase I trial by Emory University researchers in collaboration with Micron Biomedical, Inc., documenting its safety and tolerability in healthy pediatric participants (age range: 6 months–2 years) [22]. This trial utilized microneedles made of water-soluble excipients that are similar to our fabricated maltose-based dissolvable microneedle composition. In this phase 1 trial, the dissolving microneedles did not cause any adverse events in the majority of the participants, and none of them reported serious adverse events (SAEs). Out of the 33 trial participants, only grade 1 (mild) erythema, itching, and tenderness were reported (one case each). Our present trial protocol will further explore the efficacy and safety of this dissolving microneedle technology for pain alleviation in the pediatric setting.

Since pain is a subjective perception, assessment of it is difficult, especially in a pediatric population, which may eventually be a barrier to the conduct of our research. Several subjective pain assessment tools have been considered, and each possesses specific strengths and limitations [23,24]. 

Self-assessment scales require certain levels of comprehension skills and cognitive function. Hence, self-assessment scales are only reliable in children aged six years and above [25]. The Visual Analog Scale (VAS) is considered to be the most validated tool and correlates positively with other self-measuring pain scales, such as the Faces Pain Scale—Revised (FPS-R) and the Numerical Rating Scale (NRS). It has been further proven to be an optimal tool for pain evaluation, as its scores can be utilized to monitor pain progression. In addition, VAS scores are also sensitive to treatment effect, specifically when it is used to represent real differences in pain intensity at two different time points [26]. Nevertheless, recent technological advancements have developed an objective pain-monitoring device, Pain Monitor™ (Med-Storm Innovation AS, Oslo, Norway), based on enhanced skin electrical conductance associated with stress-/pain-induced sweating [27]. The device relies on the firing rate of the nociceptive nerve, mirrored by the skin conductance peaks, which can be considered the magnitude of the perceived pain stimuli [27]. Stress-induced sweating is activated through the skin sympathetic nerve through the activation of muscarinic receptors by acetylcholine and is independent of the external environment [28]. This electrical signal is then conducted and recorded by the Pain Monitor™ (Med-Storm Innovation AS, Oslo, Norway) machine as the skin conductance algesimeter index, which is the number of peaks per second produced in response to the painful stimuli.

So far, the correlation between the VAS score and skin algesimeter index has not been adequately documented in the pediatric population undergoing regular intravenous cannulation procedures for blood transfusion. Therefore, a research endeavor is required to address gaps in the current medical literature.

### 1.2. Objectives

Our general research objective is to investigate the combined efficacy of a new maltose-based dissolvable microneedle patch and the standard topical EMLA^®^ cream against the standard topical EMLA^®^ Cream with the polyvinyl alcohol (PVA)-containing polyethylene terephthalate (PET) sham patch for the provision of skin anesthesia in pediatric patients requiring intravenous cannulation for regular blood transfusion. The PVA-containing PET sham patch was chosen as a comparator since it has similar structure and characteristics to the maltose-based dissolvable microneedle patch without the micropore-forming capability of the latter, hence aiding in the blinding of the trial participants to the type of intervention received in each intervention sequence. 

We hypothesized that the use of maltose-based microneedles might reduce the dosages and onset of action required for EMLA^®^ when it is applied to the epidermal layer of pediatric thalassemia patients. Hence, we are hopeful this study may provide an improved topical analgesic method in the care of pediatric patients that require frequent intravenous cannulations or blood transfusions. 

The specific objectives of this trial are summarized as follows:(1)To compare VAS pain scores from an individual receiving the following four treatment interventions (a) using a microneedle with 1 Finger Tip Unit (FTU) EMLA^®^ cream applied for 30 min, (b) a microneedle with 0.5 FTU EMLA^®^ applied for 30 min, (c) a microneedle with 1 FTU EMLA^®^ applied for 15 min, and (d) 1 FTU EMLA^®^ cream only (without MN) applied for 30 min (control) in a randomized fashion over four consecutive visits.(2)To compare skin conductance algesimeter indices from individuals receiving the above-allocated interventions.(3)To determine the agreement between the VAS pain scores and the skin conductance algesimeter index.

## 2. Materials and Methods

### 2.1. Study Design

The design is a prospective, phase II, single-center, single-blind, cross-over, exploratory, randomized, negative-controlled (without microneedle (MN)) trial with a 1:1:1:1 allocation ratio. The trial protocol was written in line with the 2013 Standard Protocol Items: Recommendations for Interventional Trials (SPIRIT-2013) guideline [29]. This trial received ethical clearance from the UKM Research Ethics Committee (Human) (JEPUKM) (reference no: UKM PPI/111/8/JEP-2021-578; date: 19 August 2021) and has been registered with ClinicalTrials.gov (reference no.: NCT05078463) with the uploaded document dated 12 November 2021 and the Malaysian National Medical Research Registry (NMRR) (reference no: NMRR-ID-21-01989-47G). The eligible participants will be randomized in a cross-over fashion to one of 24 treatment sequences, as shown in Figure 1 and Table 1. The flowchart of this clinical trial is shown in Figure 2, and the planned schedule of patient enrolment, follow-up, data analysis, and publication of results is displayed in Figure 3.

### 2.2. Study Setting

This trial will be conducted at the pediatric daycare ward of Hospital Canselor Tuanku Muhriz (HCTM), a tertiary academic hospital under Universiti Kebangsaan Malaysia, Kuala Lumpur campus.

### 2.3. Sample Size Calculation 

The determination of sample size was carried out using G*Power software, version 3.1.9.6 (Universitat Kiel, Germany; February 2020), using the F-test family, repeated measure (RM) ANOVA within and within-between interaction sub-options. The type I error (α) and the study power (1 − β, where β = type II error) were fixed at 0.05 and 0.80, respectively.

Due to the scarcity of information from previous literature on the pediatric population, we could not use the parameter estimates (standard deviation of the differences, mean differences between pairs) from prior studies as guides for our sample size calculation. However, to circumvent this, we utilized Cohen’s guideline for choosing this study’s effect size [30]. We assumed that the Cohen’s d was large (d = 0.8) and used this as our effect size. It had been demonstrated by a previous study on the efficacy of microneedle as a delivery method for 5% lidocaine dental gel (Septodent, UK) that the effect size could be that large, thus justifying our choice of effect size in general [31]. 

For objectives 1 and 2, we used the RM-ANOVA within-group comparison sub-option and fixed the number of groups and measurements to 1 and 4, respectively. Based on prior guidelines, moderate-to-high correlations between the within-group measurements (r = 0.7) were assumed [32,33]. For non-sphericity correction (ε), a value of 0.75 was assumed since the sphericity assumption is rarely fulfilled in practical and actual trial settings due to unequal correlations between measurement pairs and the chosen ε value was midway between its lower bound of 0.5 and maximal value of 1 to fairly preserve the RM-ANOVA test against inflated type I error rate (α) when non-sphericity occurs [34,35,36,37]. Hence, for objectives 1 and 2, we require a sample size of 23 subjects. Assuming a 10% attrition (drop-out) rate, the total sample size is 26 (n_total_ = 26).

For objective 3, we could not calculate the sample size since we did not have the estimate of the standard deviation of the differences (denoted by (s) henceforth) between the two methods for pain evaluation (the VAS score and the skin conductance algesimeter index obtained via PainMonitor™). Information on (s) was required since the sample size formula for evaluating limit of agreement required parameters to be explicitly specified. However, we could roughly assess the precision of the limit of agreements (as per (s) magnitude) based on the sample size proposed for objectives 1 and 2 using the formula recommended by the original authors [38]. Based on Bland and Altman (1986), the 95% CI of limit of agreement is given by:±1.96 ∗ sqrt(3/n) ∗ s(1)
where s = the standard deviation of the differences between the VAS score and the skin conductance algesimeter index. 

If we assume a sample size of 24, then the precision of the limit of agreement is:±1.96 ∗ sqrt(3/26) ∗ s = ±0.67 ∗ s(2)
which we deemed satisfactory in our case.

However, due to the absence of an estimate of (s) from prior studies, the actual precision could not be accurately estimated, thus highlighting the importance of (s) in the actual sample size calculation. Thus, it is hoped that the estimate of (s) obtained from this study can be used as a guide for calculating the sample size for a future follow-up study that can answer this study’s objective more reliably. 

Hence, the biggest overall sample size required for this study was 26 (n_total_ = 26).

### 2.4. Recruitment

The recruitment will be carried out by screening the patient list in the pediatric thalassemia patient database. There is a pool of approximately 80 patients with blood-transfusion-dependent thalassemia who receive their blood transfusion at the daycare ward on a monthly basis. We anticipate that 30% of them will not meet the inclusion criteria, and 20% will refuse to participate in the trial. We estimate that around 90% of the patients who are eligible and recruited will complete all of the planned follow-up visits. In this context, we anticipated a study period of at least nine months for the required sample size of study participants to be enrolled until the completion of all assigned interventions before the ethics expiry date. No financial incentives will be provided by the study investigators to facilitate and enhance participant recruitment during the enrollment period.

### 2.5. Study Population and Sampling Method

Thalassemic patients who come for regular blood transfusion at HCTM who fulfill the inclusion and exclusion criteria will be recruited to the study upon the parent/guardian voluntarily providing written informed consent and, when possible, the child assenting to participate (Appendix A). Purposive sampling will be used to obtain the study subjects to prevent insufficient recruitment caused by the paucity of eligible study participants.

### 2.6. Inclusion Criteria

The inclusion criteria include (i) patients aged at least 6 years to 18 years old, and (ii) patients requiring venous cannulation for blood transfusion. 

### 2.7. Exclusion Criteria

The exclusion criteria include (i) patients with a previous history of sensitization or allergy to EMLA^®^ cream; (ii) patients with a previous history of allergy to materials used in the study, i.e., plaster, electrodes, maltose, PVA, or PET; (iii) patients exposed to analgesic usage within 24 h prior to the procedure; (iv) generalized skin disorder or rash; and (v) agitated or fretful patients. 

### 2.8. Randomization

For random allocation, simple randomization with the random allocation rule (RAR) will be used, by which a list of random numbers will be generated in a balanced 1:1 ratio with no stratification using the randomizer package version 20.0 executed on R platform [39]. The study participants, care providers, and data handlers will be blinded to the study interventions. Only the statistician, the interventionist, and the assessor will be unblinded to the study interventions. Besides, a unique code to indicate each treatment sequence assignment will be utilized to ensure that the unintentional/intentional unblinding of one trial participant will not compromise the integrity of blinding for the rest of the study participants.

Patient intervention status will be unblinded if the participants develop serious adverse events (SAE) or suspected unexpected serious adverse reaction (SUSAR). For data analysis, unblinding will be performed, and the status of the participant’s interventions will be made accessible to the statistician.

### 2.9. Allocation Concealment Mechanisms

Allocation concealment will be performed in a double fashion (i.e., the study participants and the healthcare personnel performing the cannulation and delivering the usual hospital care). Since the central randomization method is adopted, the interventionist has to contact the statistician who generated and prepared the randomization sequence via phone call to obtain the randomization sequence allotted to a newly recruited study participant. In this fashion, the interventionist will not be able to decipher or guess the next randomization sequence that will be allocated to the next study participant. 

### 2.10. Consent and Confidentiality

All personal clinical data on the data collection sheets and patient names will be replaced by the assigned randomization numbers and Subject Identification No. (SIDNO) to protect anonymity. During the treatment period, patients or their legal representatives can withdraw their permission for the research and data collection without compromising their standard medical treatment. The interventions and follow-up of the study will subsequently be cancelled. The data that have already been collected before the withdrawal of permission will be used for the analyses based on the Intention To Treat (ITT) principle, unless it is specifically declined. 

The trial data with study participant identification numbers removed (hospital RN, identity card (IC) numbers) (SIDNO) will be made available to the public via the Harvard Dataverse repository for research data (https://dataverse.harvard.edu).

### 2.11. Research Materials

The materials used and their manufacturers in this research are listed in Table 2. 

### 2.12. Fabrication of Solid Maltose Microneedles

The MN array patch size is measured at 10 mm × 10 mm, containing 36 microneedles with 1 mm needle gap in between. The maltose (Hayashibara, Okayama, Japan) MN dimension is designed to be around 400 μm in height, with a base width of 100 μm and a 3 μm tip radius. The standard deviation of needle heights within the patch is controlled to be less than 3%. The total patch thickness is therefore 0.8 mm. The MNs are grown on a soft cushion on top of PVA material (Kanto Chemical, Tokyo, Japan), with a PET patch (Acrysunday, Tokyo, Japan) to support the soft PVA patch as below. The base patch spans 125% more than the array MN patch, with an estimated batch patch size of 15 mm × 15 mm. The overall size of the MN device is thus 1.7 cm (width) × 4 cm length × 0.5 cm (height). Figure 4 illustrates the schematic representation of microneedle prototypes that will be used in this study.

The MN patch will be manually applied to the skin with normal thumb force, and various studies have reported that the hole diameter should be around 1/3 of MN length [38,39]. Therefore, this research’s specification is speculated to achieve the desired penetration depth of around 160 μm where the epidermal–dermal junction lies. To ensure a uniform force is applied all over the MN patch, the interventionist will stick a pillar handler of a size of 10 mm × 10 mm on the bottom surface of MN patch with a double cellophane tape (Sellotape^®^) stuck onto it. The application of MN to the skin mimics a stamping action.

The maltose dissolving time is about 15 s at a temperature between 25 and 30 °C, with more than 60% humidity. Hence, in normal conditions, the maltose MN can dissolve into the skin within 1 min. The maltose MN is recommended to be stored at room temperature (upper limit not exceeding 40 °C) and at a room humidity of less than 60% to avoid melting. The shelf time can last for more than 8 years, provided that the storage temperature is maintained at 25 °C and with a room humidity of less than 10%.

### 2.13. Administration of Interventions/Controls

Prior to the administration of intervention/control, relevant clinical–demographic profiles (age, gender, ethnicity, anthropometric measurements, presence of comorbidities, thalassemia types etc.) will be recorded and entered in the case report forms (CRFs) that are specifically designed for this study. This research study uses EMLA^®^ cream (lidocaine 2.5% and prilocaine 2.5%) as the topical anesthetic agent. EMLA^®^ cream is a eutectic emulsion mixture of lidocaine and prilocaine in a 1:1 ratio (i.e., each gram of EMLA^®^ cream contains lidocaine and prilocaine, 25 mg each). A eutectic mixture has a lower melting temperature than each constituent’s melting temperature. The anesthetic efficacy of EMLA^®^ cream will be assessed via pain induced by intravenous cannulation. The primary endpoint is the participant’s VAS score measured after applying EMLA^®^ cream (with and without MN application) for 15 and 30 min. 

The window period given to EMLA^®^ cream for its effect to work will be based on the usual clinical practice observation, where it is usually applied 30 min prior to intravenous catheterization. The rationale behind this is due to logistical issues and for the daycare’s operational convenience. Nevertheless, in a busy clinical setting, the application time is sometimes shortened to 15 min for a slight anesthetic effect. Thus, the study investigators postulate that, with the aid of microneedles, the time to onset of action for EMLA^®^ cream could be significantly reduced, thus requiring less time to achieve its maximal effects.

According to the routine hospital protocol, all study participants received their blood transfusion based on the Good Clinical Practice (GCP) guidelines. For each participant, the individual will be randomized to one of the four treatment sequences (Figure 1 and Table 1). Before administering the subsequent intervention, there will be a minimum of a three-week washout period. Figure 2 illustrates the complete clinical flow for the recruitment and randomization phases.

The investigator will identify and draw a grid of 1 cm × 1 cm at the dorsum hand, which serves as an ideal site for cannulation. The healthcare personnel in charge will be instructed to apply either 1 FTU of EMLA^®^ cream (approximately 0.68 g/cm^2^) or 0.5 FTU (approximately 0.369 g/cm^2^) over the identified skin area. If the patient is subjected to MN patching at his/her visit, the MN patch will be applied by thumb force and pressed firmly against the hand surface for 5 s to patch the MN to the skin entirely before applying EMLA^®^ cream. Otherwise, an empty (i.e., without MN) PVA-containing PET sham patch will be applied instead. In addition, the height-to-base ratio (4:1) used for MN will optimally minimize its adverse effects (pain, redness), thus preserving the masking (blinding) of study participants from knowing the types of interventions received. The preparation area will be covered with an adhesive dressing (Tegaderm™; 3M, Maplewood, MA, USA) after EMLA^®^ cream application. After the allocated application time (15 or 30 min), the attending healthcare personnel will set up the transfusion line with a 22-gauge intravenous cannula inserted at the dorsum of the hand. Throughout the process, the parent/guardian will be allowed to stay by the patient’s side as necessary.

### 2.14. Pain Assessment 

After a random treatment sequence is assigned, the study participants will be instructed on how to operate the 10-point, 100 mm VAS pain score. 

Before applying the MN patch and EMLA^®^ cream, the patients will be connected to the PainMonitor™ (Med-Storm Innovation AS, Oslo, Norway) device, whereby the electrodes will be attached to the hypothenar eminence of the opposite hand to the one to be cannulated for blood transfusion. The skin conductance peaks (in microSiemens (μS) and the average rise time (in microSiemens per second (μS/s)) will be recorded. 

After each procedure, the children will be asked to place the slider in the slot that accurately describes his/her pain at the following time points: (1) 1 min after application of the MN/sham patch and before EMLA^®^ cream application (baseline VAS score), and (2) 1 min after IV cannulation. The investigator will record the location of the slot where the slider is placed in millimeters (mm), and this will be the participant’s VAS score. Throughout the process, there will be a trained nurse standing by to aid the participants who require additional assistance.

All data will be analyzed at the end of the trial. Hence, no interim analysis or early stopping guidelines or decisions are applicable for this trial.

### 2.15. Strategies to Improve Adherence to Interventions

The participants will be instructed to avoid vigorous movements to ensure the EMLA^®^ patch does not accidentally peel off. 

### 2.16. Relevant Concomitant Care Permitted or Prohibited during the Trial

The participants will not be allowed to take any analgesic medications (NSAID, opioid, paracetamol etc.) during the trial day since they will modulate the level of pain experienced by the participants due to the received interventions. Other medications and concomitant care are permitted during the trial.

### 2.17. Provision of Post-Trial Care

Apart from the two phone calls at 24 and 48 h after the intervention to obtain information on the intervention-associated adverse events, no further provision of post-trial care will be instituted. The UKM Research Ethics Committee (Human) has waived the need to take insurance for study subjects due to the negligible risks associated with the interventions (microneedle, EMLA^®^, and sham patch) that will be administered in this trial.

### 2.18. Research Outcomes

The outcomes of this study will be observed in two prospects, in the clinical trial settings and data analysis outcomes. In the clinical trial settings, the observed outcomes include VAS scores and skin conductance peaks. Meanwhile, the data analysis outcomes include agreement between the Visual Analogue Scale pain scores obtained and the skin conductance algesimeter index obtained via the PainMonitor™ machine.

### 2.19. Data Collection

The data will be collected for all the independent and dependent variables of the patient. The recorded data will be recorded on a paper-based case report form that is available as Appendix A. Non-numerical data will be coded using standardized code to facilitate data storage, review, and analysis. The data will also be checked for accurate format and that the values fall under the anticipated range of values.

Independent variables: (i) Age: The age of a study participant at the first study visit. The variable will be measured in years and months and modeled as a continuous numerical variable and will not be categorized into separate age groups. (ii) Gender. (iii) Ethnicity. (iv) Body mass index (BMI): a continuous numerical variable that is calculated using the standard body mass index formula; BMI = kg/m^2^. This variable will be categorized according to the WHO BMI classification [40]. (iv) Intervention groups. (v) Baseline VAS score: This will be used as a predictor variable to control the confounding effect of heterogeneous baseline VAS scores among study participants. (vi) Baseline pain score obtained via the PainMonitor^™^ device: This will be used as a predictor variable to control the confounding effect of heterogeneous baseline VAS scores among study participants.

Dependent (outcome variables): (i) VAS score (15 and 30 min post EMLA^®^ application during IV cannulation): a continuous numerical variable that will be measured during each visit, 15 min after EMLA^®^ application and 30 min after EMLA^®^ application. (ii) Pain score from the PainMonitor^™^ device (15 and 30 min post EMLA^®^ application): a continuous numerical variable that will be measured during each visit, 15 min after EMLA^®^ application and 30 min after EMLA^®^ application.

The monthly visits will be scheduled by the respective healthcare provider. The outcome data will not be collected from any non-retention or non-adherent participants.

### 2.20. Data Management 

The data will be recorded on the case report form. Non-numerical data will be coded using standardized code to facilitate data storage, review, and analysis. The data will also be checked for proper format and whether the values fall under the anticipated range of values. The accuracy of data entry will be further examined using the double data entry approach and during exploratory data analysis.

The auditing process will not be required in this trial since the progress of this trial will be monitored by the independent UKM Research Ethics Committee (Human) and the Malaysian Ministry of Science, Technology and Innovation (MOSTI).

### 2.21. Statistical Analysis 

All data will be descriptively summarized as mean and standard deviation (SD) for the normally distributed numerical variables and frequency and percentage for categorical variables. For numerical data, the normality assumptions will be evaluated subjectively using the histogram with an overlying normal distribution curve, and objectively using the Kolmogorov–Smirnov and Shapiro–Wilks tests (*p* > 0.05 indicates the normal distribution assumption is met). Fisher’s coefficient of skewness will be subsequently utilized to assess the severity of skewness in non-normally distributed numerical variables. In addition, repeated-measure ANOVA and the generalized mixed-effect model (GLMM) with an identity-link function will be used to assess and estimate the differences in terms of VAS score means between the periods of interventions, considering intra-subject correlations by using the restricted maximum likelihood (REML) estimator. Moreover, the multiple imputation method (MICE) will also carry out sensitivity analyses to investigate the effects of missing data under the missing-at-random (MAR) framework. In order to evaluate the correlation between the VAS pain score and the PainMonitor™ device, the Pearson correlation coefficient, Spearman coefficient, intraclass correlation coefficient (ICC), and Bland and Altman (B&A) plot will be used to assess the agreement between the two scores.

For each analysis, the significant threshold for the *p*-value is fixed at 0.05, and the 95% confidence interval (CI) will be obtained for each parameter estimate. Data analysis will be performed using Statistical Package for Social Science (SPSS™) (IBM Corp. Released 2020. IBM Statistics for Windows, Version 27.0, Armonk, NY, USA: IB Corp).

### 2.22. Ethical Issues

Voluntary written informed consent will be obtained from the parent or legal representative of each study participant. This study will be conducted in accordance with the principles of ethics in human research as stipulated by the Declaration of Helsinki (64th World Medical Association General Assembly, 2013) and the Good Clinical Practice (GCP) guidelines [41,42]. Ethical approval has been obtained from the UKM Research Ethics Committee (JEPUKM) (UKM PPI/111/8/JEP-2021-578; date: 19 August 2021).

To aid the transparency of reporting, the trial has been registered at the Clinical Trials.gov registry (NCT05078463). The full trial protocol will be made available in the same trial registry.

### 2.23. Trial Oversight and Monitoring

#### 2.23.1. Data Monitoring Committee

Data monitoring and quality assurance are guaranteed by an inspection of all CRFs together with other documents such as the clinical report file and the discharge letter issued after the completion of blood transfusion during the daycare visit by the two research assistants (RAs) and statistician responsible for this project. Moreover, several subject CRFs and informed consent forms will be extensively reviewed, which include thorough reviews on eligibility criteria, completeness of data, and details recorded on the CRFs and informed consents forms; pseudoanonymization of trial data; and data entry on the SPSS spreadsheet on a monthly basis. The progress of data collection and overall trial conduct will be relayed in writing to the Data Monitoring Committee (DMC) under the UKM Research Ethics Committee every six months.

#### 2.23.2. Safety Assessment

The participants will be allowed to withdraw from the study at any time and for any reason. Standard local clinical practice will be instituted to cater to the participant’s need in the event of study withdrawal.

Study protocol will be halted at any moment of the study period if the study participants develop any sudden (expected or unexpected) severe complications/adverse events. The withdrawal of a study participant from the trial will be documented on the adverse event page of the CRF and the participant will be further followed up for the study outcomes and included in the analysis as per the original randomization group (intention-to-treat analysis). All AEs/SAEs will be recorded and graded according to the Common Terminology Criteria for Adverse Events (CTCAE) version 5 and the US FDA’s Toxicity Grading Scale Healthy Adults and Adolescents Volunteers Enrolled in Preventive Vaccine Clinical Trials criteria [43,44]. 

#### 2.23.3. Plans for Communicating Important Protocol Modifications

Any protocol amendments will be notified by letter to the UKM Research Ethics Committee (Human) for approval. The principal study investigators will submit documentation related to the progress of the trial to the funder (the Malaysian Ministry of Science, Technology and Innovation Prototype Research Grant Scheme (MOSTI-PRGS)) and the UKM Research Ethics Committee (Human) every six months. The details include the date of the recruitment of the first study participant, the number of recruited participants, the number of participants who have completed all trial interventions, and any documented AEs, SAEs, or SUSARs.

#### 2.23.4. Dissemination Policy

The publication of the results for this clinical trial will be initiated by the principal investigators and all other co-investigators of this trial. The results will be published in peer-reviewed journals and presented at international conferences of reputable esteem. The funder, the Malaysian Ministry of Science, Technology and Innovation (MOSTI) under the Prototype Research Grant Scheme (PRGS), also has the right to publish the research objectives, study plans, and cost associated with conducting this clinical trial to the public.

The full trial protocol and statistical analysis plan are made available to the public at the ClinicalTrials.gov registry (https://www.clinicaltrials.gov/ct2/show/NCT05078463). The statistical programming code for data analysis and the randomization procedure will be made available to the public as appendices to this publication.

## 3. Discussion

This trial protocol studies pain response to intravenous cannulation preceded by application of topical anesthetic with or without MN, in the pediatric population with thalassemia requiring regular blood transfusions. By using a microneedle patch with topical anesthetic, it is hypothesized that this combination may accelerate relief of pain caused by the cannulation process.

On the other hand, the challenges to the successful completion of the study are the variability of pain score numerical concepts among the pediatric patients and the compliance of the patient. Thalassemic patients may have skin thickening from increased iron deposition that may affect the cannulation process and pain thresholds. Objectively, the protocol requires patients to rate the pain after the procedure, and the pediatric patient may not have a clear concept or understanding of pain scores, which causes a variability of pain scores in different patients. In addition, patient compliance might be a challenge due to the cross-over study design that requires patients to comply with the clinical trial protocol. The additional time taken may be a hassle to the trial participants receiving care at their monthly blood transfusion routine at the pediatric daycare ward. However, this is mitigated by clear instruction and counseling on the trial protocol and commitments agreed upon and ensuring the same healthcare personnel supervising the procedures, thus minimizing participants’ inconvenience.

One of the main strengths of this trial is, in comparison to the previous studies using microneedles, that the study is specifically conducted in a pediatric population. The same individual becomes his/her internal control with a washout period of 3–4 weeks as an interval between the allocated four intervention arms, also coinciding with the patient blood transfusion visits. So far, there is no study that has investigated the efficacy of a dissolvable microneedle technology as adjunct with topical anesthetic application for pain relief in a pediatric population requiring regular intravenous cannulation. Besides, the fabrication technology is maltose-based, which may further provide a safe (minimal adverse effects), stable, and efficacious patch to ease cannulation-associated pain. Furthermore, the pain score will be measured objectively (pain machine) and validated against a standard VAS score.

## 4. Trial Status

The trial recruitment is based on protocol version 2.0. The first patient was recruited and enrolled in this trial in September 2021, and the last patient was recruited in February 2022.

## Figures and Tables

**Figure 1 jcm-11-05291-f001:**
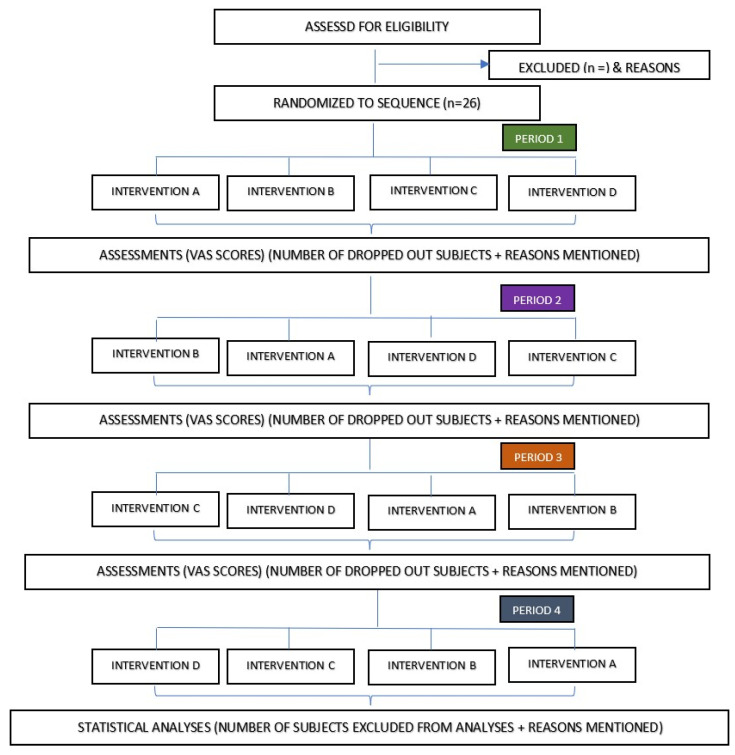
The flowchart of this randomized cross-over trial shows four random intervention sequences out of 24 permutations generated. Intervention A: MN + 1 FTU EMLA^®^ 30 min application; intervention B: MN + 0.5 FTU EMLA^®^ 30 min application; intervention C: MN + 1 FTU 15 min application; intervention D: 1 FTU EMLA^®^ only 30 min (controls).

**Figure 2 jcm-11-05291-f002:**
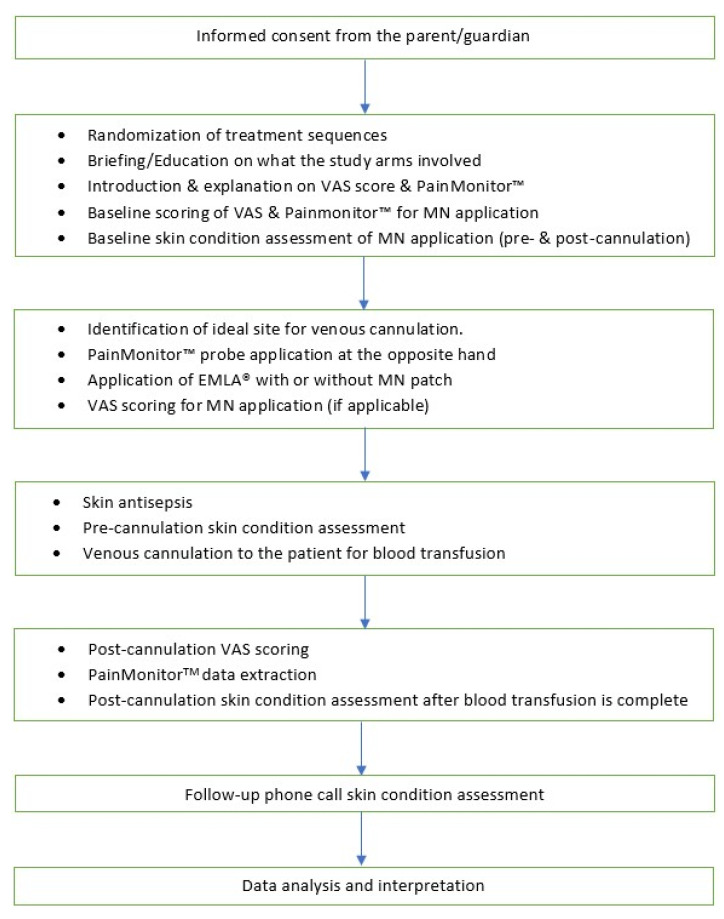
Schematic flow of the clinical trial.

**Figure 3 jcm-11-05291-f003:**
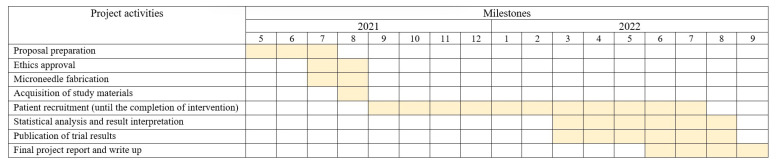
Milestone chart of the clinical trial.

**Figure 4 jcm-11-05291-f004:**
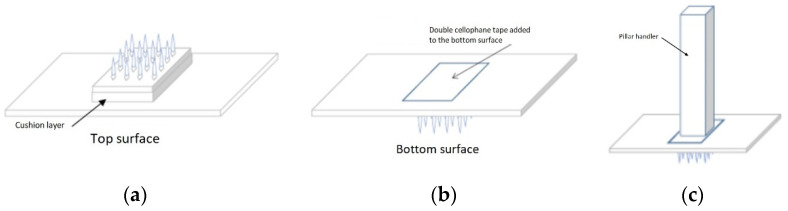
The schematic representation of microneedle prototypes. (**a**) The top surface of the microneedle patch. (**b**) The bottom surface of the microneedle patch. (**c**) The final design of the dissolvable maltose-based microneedle patch.

**Table 1 jcm-11-05291-t001:** The list of permutations of intervention sequences randomized for the trial participants. Intervention A: MN + 1 FTU EMLA^®^ 30 min application; intervention B: MN + 0.5 FTU EMLA^®^ 30 min application; intervention C: MN + 1 FTU EMLA^®^ 15 min application; intervention D: 1 FTU EMLA^®^ only 30 min (controls).

ABCD	BACD	CADB	DABC
ACBD	BADC	CABD	DACB
ABDC	BDAC	CBAD	DBAC
ACDB	BDCA	CBDA	DBCA
ADBC	BCAD	CDAB	DCAB
ADCB	BCDA	CDBA	DCBA

**Table 2 jcm-11-05291-t002:** Research materials and their manufacturers used in this research.

Materials	Manufacture
PainMonitor™	Medstorm, Norway
EMLA^®^ Cream	Aspen, Malaysia
Microneedle patch	Alnair Photonics, Japan
Sham patch	Alnair Photonics, Japan
Visual Analogue Scale ruler	Schlenker Enterprises, Illinois, USA

## Data Availability

The data from this study are available on request from the corresponding author. The data are not publicly available due to patient confidentiality and are used under license for the current study.

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
