# Peer review of "Transdermal Maltose-Based Microneedle Patch as Adjunct to Enhance Topical Anesthetic before Intravenous Cannulation of Pediatric Thalassemic Patients Receiving Blood Transfusion: A Randomized Controlled Trial Protocol"

_jcm, 2022, doi:10.3390/jcm11185291_

Round 1

Reviewer 1 Report

MS Title: Intravenous cannulation is experientially traumatic to children….

MS No: jcm-1827318

MS authors: Muhammad Irfan Abdul Jalal et al.

This is a prospective randomized cross-over control trial. The protocol presents the purposes and methods to investigate the possible effect of maltose microneedle on the analgesic action of EMLA ointment in pediatric thalassemia patients who need regular intravenous cannulation and blood transfusion. The study protocol was written according to the GCP requirements and in compliance of ethics principles in Declaration of Helsinki.

Minor issues:

1.      Page 1, line 33: “The trial will commence in August 2021 and is anticipated to end by August 2022.” It seems that the trial has started September 2021 and recruited the last subject in February 2022. Therefore, I am not sure whether the tense of the verbs in this protocol article should be in future verb tense or past tense.

2.      Page 4, line 150: “ftu” should be “FTU”

3.      Page 7, line 222: “± 1.96* sqrt(3/n)*s37” better to remove “37”

4.      Page 8, 238: “we estimate that around 90 % of the patients who show clinical equipoise.” How to reach this estimation?

5.      Page 8,line 250: “inclusion criteria” Is it possible to know how many trial subjects had painful IV cannulation for transfusion experience before entering this trial?

6.      Page 9, line 270: “SUSAR” should be “suspected, …….”

7.      Page 14, line 461: “the Declaration of Helsinki (18th World Medical Association General Assembly, 1964"better to cite “64th WMA General Assembly, Fortaleza, Brazil, October 2013”

8.      Page 14, line 462: “Good Clinical Practice (GCP) guidelines.31” The reference 31 ??

9.      The informed consent form:

(a)    “Voluntary written informed consent will be obtained from each study participant."The consent will be obtained always from trial subject’s LAR (line 719: your child)? Any assent from the pediatric subjects? Line 772: “you or a legal representative”!

(b)   Readability: for 6 years or above, e.g., “non-invasive” , “cannulation”, “anesthesia”, “clinical setting”, “microneedle”, “superficial level”

(c)    Any version of ICF written in own language instead of English?

10.   Any information provided by phase 1 trial results?

Author Response

Thank you for your comments. Our responses to your queries and the amendments made are as follows:

  1. Page 1, line 33: “The trial will commence in August 2021 and is anticipated to end by August 2022.” It seems that the trial has started September 2021 and recruited the last subject in February 2022. Therefore, I am not sure whether the tense of the verbs in this protocol article should be in future verb tense or past tense.

Response:

We believe that the sentence should be in future tense since this protocol was devised, finalized and registered with the clinical trial registry before any single patient recruitment was carried out.

  1. Page 4, line 150: “ftu” should be “FTU”

Response:
Thank you for pointing this out. The correction has been made accordingly.

  1. Page 7, line 222: “±1.96* sqrt(3/n)*s37” better to remove “37”

Response:

Thank you for pointing this out. The superscript (37) has been removed accordingly.

  1. Page 8, 238: “we estimate that around 90 % of the patients who show clinical equipoise.” How to reach this estimation?

Response:

We have amended the term "clinical equipoise" to the following for better clarity. Our estimation is based on the experience gained while managing these patients in the pediatric daycare setting here, thus the new sentence is (Pg 9, from Line 255, highlighted in yellow):

We estimate that around 90 % of the patients who are eligible and recruited will complete the whole planned follow up visits. In this context, we anticipate a study period of at least nine months for the required sample size of study participants to be enrolled until the completion of all assigned interventions before the ethics expiry date.

  1. Page 8,line 250: “inclusion criteria” Is it possible to know how many trial subjects had painful IV cannulation for transfusion experience before entering this trial?

Response:

As stated in our response to Q4 above, based on our experience and the pool of patients we have, we could estimate the number eligible and the anticipated study sample size to be recruited and the completion of the study over a period  of about nine months before the ethics expiry date.

  1. Page 9, line 270: “SUSAR” should be “suspected, …….”

Response:
Thank you for pointing this out and apologies for this typing error. The correction has been made accordingly.

  1. Page 14, line 461: “the Declaration of Helsinki (18th World Medical Association General Assembly, 1964"better to cite “64th WMA General Assembly, Fortaleza, Brazil, October 2013”

Response:
Thank you, we have revised the citation accordingly.

  1. Page 14, line 462: “Good Clinical Practice (GCP) guidelines.31” The reference 31 ?

Response:

Apologies for this typing error. The reference number has been corrected accordingly to 41-42.

  1. The informed consent form:

(a)    “Voluntary written informed consent will be obtained from each study participant."The consent will be obtained always from trial subject’s LAR (line 719: your child)? Any assent from the pediatric subjects? Line 772: “you or a legal representative”!

(b)   Readability: for 6 years or above, e.g., “non-invasive” , “cannulation”, “anesthesia”, “clinical setting”, “microneedle”, “superficial level”

(c)    Any version of ICF written in own language instead of English?

Response:

a) We have amended the sentence (Pg 15, Line 487, highlighted in yellow) from “Voluntary written informed consent will be obtained from each study participant.” to “Voluntary written informed consent will be obtained from the parent or legal representative of each study participant”.

With regards to the sentence “you or legal representative” (Line 772), the word “you” refers to the parent or a legal representative of the child.

Where possible, we also obtain assent from the child to participate in this study.

With regards to “Signature of Patient or Legal Representative” (line 805):

Apologies, this was from an older version. In the updated version which was in the actual consent form used, it reads “Signature of Parent or Legal Representative”.

This consent form, revised version 2.0 of our clinical trial protocol is re-uploaded in the supplementary. 

b) This information sheet is given to the parent/guardian who are expected to understand these terms. Furthermore, for each terminology, the person who took the consent from the parent/child’s legal representative also explained the meaning of each technical word according to parent’s level of understanding and educational status. 

c) There are two versions of the ICF. Based on your query, we have included and uploaded the local language (Bahasa Malaysia) version of the ICF in the supplementary.

10. Any information provided by phase 1 trial results?

Response:

We have referred to the results of a phase I study sponsored by Emory University with Micron Biomedical, Inc as the industrial collaborator (Clinicaltrials.gov ID:  NCT03207763, link: https://clinicaltrials.gov/ct2/show/study/NCT03207763). We have included this mention in the research background section 1.1, Pg 3, Lines 107-120, highlighted in yellow).  This study used microneedles made of water-soluble excipients that are similar to our microneedle composition. The majority of the participants did not report any adverse events and none of the subjects reported any serious adverse events (SAEs). Out of 33 participants in that RCT, one participant reported grade 1 (mild) erythema,  itching and/or tenderness.  In this regard, the safety and tolerability of the dissolvable microneedle technology has been ascertained to some extent by this study, which will be further explored in our present study.

Reviewer 2 Report

The planned study addresses a clinical problem for pediatric venous cannulation. Prevention of cannulation pain is of great importance regarding compliance in children. In case that the addition of microneedling will improve local analgesia with EMLA cream it could be very helpful in daily practice.

The presented study protocol is clearly and comprehensively written. A weakness of the presented protocol is that it was not published earlier in the course of the study but it seems still worth publishing to make all details available.

I have an important comment:

The study protocol was changed at the beginning of the trial. This is version 2.0. Please state the changes, the reason, and how many patients were recruited until then, or were all patients excluded prior to the start of version 2.0?

Author Response

We thank the reviewer for your kind comments and appreciation.

Query:

The study protocol was changed at the beginning of the trial. This is version 2.0. Please state the changes, the reason, and how many patients were recruited until then, or were all patients excluded prior to the start of version 2.0?

Response:

Version 2.0 of the study protocol was the final version that was adopted for the trial. This version utilized a refinement in the dimension of the microneedle  from 2 cm x 2cm (total area: 4cm2) to 1 cm x 1 cm (total area: 1 cm2)

taking into account a better operational functionality

The section on interventional safety assessment had been further expanded and elaborated.  

There were no patients recruited using protocol version 1, as recruitment only started after version 2.0 was completed as the final protocol. Hence, no patients were excluded prior to the start of version 2.0. The version number was merely for our own internal reference.